# miR403a and SA Are Involved in *NbAGO2* Mediated Antiviral Defenses Against TMV Infection in *Nicotiana benthamiana*

**DOI:** 10.3390/genes10070526

**Published:** 2019-07-12

**Authors:** Pengfei Diao, Qimeng Zhang, Hongyu Sun, Wenjie Ma, Aiping Cao, Ruonan Yu, Jiaojiao Wang, Yiding Niu, Hada Wuriyanghan

**Affiliations:** Key Laboratory of Forage and Endemic Crop Biotechnology, Ministry of Education, School of Life Sciences, Inner Mongolia University, Hohhot 010070, China

**Keywords:** *Nicotiana benthamiana*, *NbAGO2*, antiviral defense, miR403a, SA, TMV

## Abstract

RNAi (RNA interference) is an important defense response against virus infection in plants. The core machinery of the RNAi pathway in plants include DCL (Dicer Like), AGO (Argonaute) and RdRp (RNA dependent RNA polymerase). Although involvement of these RNAi components in virus infection responses was demonstrated in *Arabidopsis thaliana*, their contribution to antiviral immunity in *Nicotiana benthamiana*, a model plant for plant-pathogen interaction studies, is not well understood. In this study, we investigated the role of *N. benthamiana NbAGO2* gene against TMV (Tomato mosaic virus) infection. Silencing of *NbAGO2* by transient expression of an hpRNA construct recovered GFP (Green fluorescent protein) expression in *GFP*-silenced plant, demonstrating that *NbAGO2* participated in RNAi process in *N. benthamiana*. Expression of *NbAGO2* was transcriptionally induced by both MeSA (Methylsalicylate acid) treatment and TMV infection. Down-regulation of *NbAGO2* gene by amiR-*NbAGO2* transient expression compromised plant resistance against TMV infection. Inhibition of endogenous miR403a, a predicted regulatory microRNA of *NbAGO2*, reduced TMV infection. Our study provides evidence for the antiviral role of *NbAGO2* against a *Tobamovirus* family virus TMV in *N. benthamiana*, and SA (Salicylic acid) mediates this by induction of *NbAGO2* expression upon TMV infection. Our data also highlighted that miR403a was involved in TMV defense by regulation of target *NbAGO2* gene in *N. Benthamiana*.

## 1. Introduction

Virus diseases threaten crop production and agriculture globally. Plant evolved and developed sophisticated mechanisms to defend against virus infections. RNA interference (RNAi) is an antiviral defense mechanism in plants. The history of our understanding of RNAi for viral defense can be traced back to 1928, when Wingard reported the recovery of virus infected plant from later virus infection [1]. RNAi phenomenon was then reported in petunia in 1990 and was termed as “co-suppression” and later as post transcriptional gene silencing (PTGS) [2]. In 1993, Lindbo et al. suggested a working mechanism for RNAi while studying gene silencing phenomena in transgenic plants [3]. In 1998, Fire et al. silenced endogenous gene expression in *Caenorhabditis elegans* by injection of double-stranded RNA (dsRNA), which was therefore identified as the molecule mediating sequence specific inhibitory effects during RNAi process [4]. In 1999, Hamilton and Baulcombe found that 25-nucleotide antisense RNA served as the intermediate for RNAi [5]. RNAi was also found in fungus and virus infected plant, and was coined as quelling and virus-induced gene silencing (VIGS) respectively [6,7,8]. Briefly, RNAi mechanism is as follows: dsRNA of different sources are excised by the RNase III family endoribonuclease Dicer/DCL (Dicer-like) to generate small interfering RNA (siRNA) with the length of approximately 21–24 nucleotides. The guide strand, which is complementary to target RNA, is incorporated into the RNA-induced silencing complex (RISC), the main component of which is the PIWI family Argonaute (AGO) protein. RISC-incorporated siRNA then finds the target RNA by base-pairing and degrades it by the activity of AGO protein. RNA dependent RNA polymerase (RdRp, RDR) amplifies the silencing signal via the synthesis of secondary siRNAs from primary single-stranded siRNAs [9].

Although both AGO and Dicer/DCL proteins are involved in RNAi process, their functions are diverged in defense responses against different viruses. Furthermore, there are many AGO and DCL members in plants and they showed specialized roles against different pathogens. In model system *Arabidopsis thaliana*, the functions of these two family proteins were heavily investigated. AGO1 plays key role against viruses like turnip crinckle virus (TCV, *Tombusvirus*) [10], cucumber mosaic virus (CMV, *Cucumovirus*) [11], potato virus X (PVX, *Potexvirus*) [12] and turnip mosaic virus (TuMV, *Potyvirus*) [13]. AGO2 is involved in antiviral role against TuMV [14], TCV, CMV [15] and PVX [16]. AGO4 functions against PVX [17], CMV [18], beet curly top virus (BCTV, *Geminivirus*) [19] and *Plantago asiatica* mosaic virus (PlAMV, *Potexvirus*) [20]. DCL1 is involved in antiviral role against nuclear DNA viruses like cauliflower mosaic virus (CaMV, *Caulimovirus*) [21], while DCL2 plays role against TuMV [22] and TCV [23]. RDR1 is involved in antiviral role against tobacco rattle virus (TRV, *Tobravirus*) [24], TuMV [22] and CMV [25]. RDR2 contributes to plant responses to TRV [24] and TuMV [22]. In other plant species, RDR1 plays antiviral role against sugarcane mosaic virus (SCMV, *Potyvirus*) in maize [26] and against PVX and tobacco mosaic virus (TMV, *Tobamovirus*) in tobacco [27]. In rice, RDR1 and RDR6 functions against brome mosaic virus (BMV, *Bromovirus*) and rice stripe virus (RSV, *Tenuivirus*), but not wheat dwarf geminivirus (WDV, *Geminivirus*) [28]. In virological model species *Nicotiana benthamiana* (*N. benthamiana*, *Nb*), several but limited studies were reported on antiviral role of AGO and DCL family proteins. NbDCL4 plays key role against zucchini yellow mosaic virus (ZYMV, *Potyvirus*) in *N. benthamiana* [29]. Although NbRDR1 is involved in antiviral rsponses in both tobacco and *Arabidopsis thaliana*, but is not functional in *N. benthamiana* [30]. NbRDR6 plays role against TCV, PVX and TMV in *N. benthamiana* [30]. NbRDR6 is also involved in plant responses to potato virus Y (PVY, *Potyvirus*) and the Y satellite of CMV but not TRV or CMV alone [31]. AGO proteins are key players in RNA interference pathway and RNAi-mediated antiviral mechanisms. Higher plants encode many AGOs, but their functions are not fully known. The role of *NbAGO2* was broadly investigated in *Arabidopsis thaliana*, and its antiviral role in *N. benthamiana* was also reported for TBSV (*Tombusvirus*), PVX (*Potexvirus*), TuMV (*Potyvirus*) and TCV (*Tombusvirus*) [32,33]. Therefore, we investigated the role of *NbAGO2* in RNAi-mediated antiviral role against infection by TMV, a virus belonging to *Tobamoviridae*. From its first use in *Arabidopsis thaliana* in 1996, Artificial microRNA (amiRNA) has been applied in many model and monocot species including *Oryza sativa* and *Brachypodium distachyon* [34,35,36]. Virus-based expression vector was also developed for amiRNA expression [37]. We used amiRNA technology to silence *NbAGO2* expression and investigated its role for TMV responses. Our results showed that *NbAGO2* played key roles in antiviral responses against TMV in *N. benthamiana*, and miR403a and SA was implicated in these responses.

## 2. Materials and Methods

### 2.1. Plant Cultivation and Treatment

Wild type and 16c GFP (Green fluorescent protein)-transgenic (GFP16c) *N. benthamiana* plants were used. Wild type *N. benthamiana* was provided by Prof. Xiaohan Mo from the Yunnan Tobacco Research Institute (Kunming, China) and the GFP16c line was provided by Prof. Feng Li from Huazhong Agricultural University (Wuhan, China). *N. benthamiana* seeds were cultured in sterilized nutrient soil and vermiculite (1:1) at growth temperature of 22–25 °C with photoperiod of 14:10 (light:dark) and 70% relative humidity. Three-leaf stage seedlings were used for following experiments. 0.5 mM MeSA (Methylsalicylic acid) in water solution was sprayed on the leaves and the leaves were collected one day after treatment. Water solution was sprayed as mock treatment. For GFP visualization, TMV-GFP infected plants were illuminated with a hand-held UV lamp at 320 nm or under Nikon SMZ18 microscope (Nikon, Tokyo, Japan).

### 2.2. SA Content Measurement

The content of endogenous salicylic acid (SA) was determined by UV Vis Spectrophotometry. A set of standard solutions, containing 5, 10, 20, 30, and 40 μg/mL of SA, were prepared in 60% ethanol solution and one of the standard solution was used for measuring maximum absorption wavelength within a 280–320 nm wavelength range (5 nm interval), which was used as a subsequent detection wavelength. The maximum absorption wavelength was 310 nm and the linear regression equation according to the absorbance of five standard solutions was deduced. *N. benthamiana* leaves infected with TMV and mock solution were collected and frozen in liquid nitrogen after five hours. The collected leaves were ground in liquid nitrogen and soaked with 5 mL 60% ethanol, and the supernatant after centrifugation was used to detect absorbance at 310 nm, which was then converted to SA concentration according to the standard curve [38].

### 2.3. Vector Construction

#### 2.3.1. Construction of hpGFP Vector

For silencing of *GFP* gene, *GFP* interfering sequences were amplified by Forward primer
5′-TGCGGGATATCGGACGACGGGAACTACAAGA-3′
and reverse primer
5′-TGCGGGATATCAAAGGGCAGATTGTGTGGAC-3′
from genomic DNA of GFP16c line. The fragment was cloned into pQBV3 entry vector by *EcoR*V and then was subcloned into binary vector pCB2004B to produce hairpin construct for insert *GFP* fragment.

#### 2.3.2. Construction of amiR Based Silencing Vector for *NbAGO2* Gene

For construction of viral pCV (CaLCuV)-based (cabbage leaf-curl virus) amiR vector, following oligonucleotides were designed by amiR design function in www.benthgenome.com website.

amiR1:
5′-GGATCTAGATTGATCTGAAGGAGCTGAGTTGGAGGGTTTAGCAGGGTGAAGTAAAG-3′
5′-GGAGGTACCTTGATCTGAAGTCGCTGAGTAGAAGAGTGAAGCCATTAAAGGG-3′

amiR2:
5′-GGATCTAGAAAGCTAGGTGGGATAAGTCGTGGAGGGTTTAGCAGGGTGAAGTAAAG-3′
5′-GGAGGTACCAAGCTAGGTGGTCTAAGTCGAGAAGAGTGAAGCCATTAAAGGG-3′

amiR3:
5′-GGATCTAGAGACCACAGGACGCAGAAGATTGGAGGGTTTAGCAGGGTGAAGTAAAG-3′
5′-GGAGGTACCGACCACAGGACTGAGAAGATAGAAGAGTGAAGCCATTAAAGGG-3′.

The above primers were used to amplify amiR fragment using *Arabidopsis thaliana* miR159b as template backbone and the amplicons were cloned into pCVA vector by *Xba*I/*Kpn*I sites. The same amplicons were cloned into pBI121 vector by *Xba*I/*Kpn*I sites to produce non-viral binary vector based amiR constructs.

#### 2.3.3. Construction of *NbAGO2* Overexpression Vector (Ox-NbAGO2)

To overexpress *NbAGO2* gene, open reading frame (ORF) region of *NbAGO2* was amplified by the primers
5′-TACGGATCCATGGATCGTGGAAATTACCG-3′
and
5′-TCAGAGCTCTCAGACAAAGAACATTTTGAAC-3′
from cDNA of *N. benthamiana* and the amplicon was cloned into pBI121 vector by *BamH*I/*Sac*I sites.

#### 2.3.4. Construction of STTM-Based Inhibitory Vectors for Endogenous microRNAs

For development of STTM (Short tandem target mimic) constructs for miR390a, miR393a and miR403a, the following sequences and their complementary sequences were designed and synthesized. miR390a, 5′-ctcgagGGTGCTATCCCCTATCCTGAGCTTGTTGTTGTTGTTATGGTCTAATTTAAATATGGTCTAAAGAAGAAGAATGGTGCTATCCCCTATCCTGAGCTTgaattc-3′; miR393a, 5′-ctcgagCCAAAGGGCTAATAGCATGATGTTGTTGTTGTTATGGTCTAATTTAAATATGGTCTAAAGAAGAAGAATCCAAAGGGCTAATAGCATGATgaattc-3′; and miR403, 5′-ctcgagAGTTTGTGCCTAGTGAATCTAAGTTGTTGTTGTTATGGTCTAATTTAAATATGGTCTAAAGAAGAAGAATAGTTTGTGCCTAGTGAATCTAAgaattc-3′. The restriction endonuclease sites are shown in lower-case letter, and the STTM spacer sequences were underlined. The oligonucleotides and complementary sequences were annealed to produce dsDNA and this dsDNA was then ligated into the pGreen-GUS-Competitor vector by *Xho*I and *EcoR*I respectively.

### 2.4. Plant Inoculation, Gene Silencing and Virus Infection

The above constructs were transformed into *Agrobacterium tumefaciens* GV3101 competent cells. The transformants were grown to optical density OD_600_ = 1 at 28 °C, cells were collected by centrifugation and resuspended in inoculation buffer (10 mM 2-(*N*-Morpholino)ethanesulfonic acid (MES), 10 mM MgCl_2_, 200 μM acetosyringone). OD_600_ values of agrobacterium inoculation solution were all adjusted to 1.0 unless stated otherwise. The inoculation solution was leaf-infiltrated into three-leaf stage *N. benthamiana* by needleless syringe. For pCV-based silencing experiment, pCVB was simultaneously transformed and co-infiltrated with recombinant pCVA vectors. For virus infection, infectious clone of TMV expressing green fluorescent protein (TMV-GFP), pJL24, was infected as the same way. For pJL24, OD_600_ value of *Agrobacterium* inoculation solution was 0.1. In all experiments, inoculation buffer with empty vector was used as mock treatment.

### 2.5. Total RNA Extraction and cDNA Preparation

Leaf tissues were grounded in liquid nitrogen by using mortar and pestle. The powder was homogenized in Trizol reagent (CWBio, Beijing, China) and extracted following manufacturer’s protocols. DNase was used to remove contaminated genomic DNA. Total RNA quality was confirmed by measurement of OD_260_/OD_280_ using NanoDrop2000 Spectrophotometer (ThermoFisher Scientific, Waltham, MA, USA) and agarose gel electrophoresis. One µg of total RNA was used as template and cDNA was synthesized by random hexamer using GoScript Reverse Transcription System (Promega Corporation, Madison, WI, USA).

### 2.6. Assessment of *NbAGO2* Expression by qRT-PCR Experiment and Semi-Quantitative RT-PCR

The cDNAs were subjected to PCR experiment. The PCR reaction was run on qTOWER 2.2 (Analytik Jena AG, Jena, Germany) equipment. Fast SYBR Green Master Mix (ThermoFisher Scientific, Waltham, MA, USA) was used for evaluation of *NbAGO2* level by qRT-PCR. The primer sets 5′-ATGTGAAATGGTACGGGCTGAA-3′ and 5′-CAACAAGGTTCCACTGGCATTT-3′ were used for *NbAGO2*. As an internal control, steady state level of *N. benthamiana GAPDH* gene was evaluated by primer set 5′-CTGACAAGGACAAGGCTGCT-3′ and 5′-AAGCAGCTCTTCCACCTCTC-3′. The ratio of mRNA level between treatment and mock samples was determined by the 2^−ΔΔCT^ method. At least three biological replicates with two technical replications were carried out, and the differences were evaluated by *t*-test.

### 2.7. Northern Hybridization Experiment

Ten µg of total RNA was resolved by agarose gel electrophoresis. RNAs were transferred from the gel to nylon membrane (Amersham Hybond-N^+^, positively charged) (GE Healthcare Life Sciences, Pittsburgh, PA, USA) by an electrokinetic capillary apparatus. ^32^P or digoxigenin-labelled probes were prepared by the methods described as previously, and hybridization and signal visualization were performed as per the literatures [39,40]. For *N. benthamiana NbPR1a* gene, a contig sequence with transcript ID of JN247448 from *N. benthamiana* sequence database (www.benthgenome.com) was used and the primer pairs 5′-GGATGCCCATAACACAGCTC-3′ and 5′-CCTAGCACATCCAACACGAA-3′ were used for amplification of 313bp probe sequence. For *N. benthamiana β-actin* gene, a partial cDNA sequence (JQ256516.1) was used and the primer pairs 5′-CCCAAAGGCTAATCGTGAAA-3′ and 5′-GCAGCTTCCATTCCAATCAT-3′ were used for amplification of 483bp probe sequence. For TMV detection, nucleotide encoding capsid protein was used and the primer pairs 5′-CAAGCTCGAACTGTCGTTCA-3′ and 5′-GACCAGAGGTCCAAACCAAA-3′ were used for amplification of 352bp probe sequence.

## 3. Results

### 3.1. Transient Silencing of *NbAGO2* in N. benthamiana by amiR-*NbAGO2*

Artificial miRNA recently emerged as a newly developed technology for gene silencing. *NbAGO2* amiR sequences were designed in www.benthgenome.com website and were used to amplify amiR fragment using *Arabidopsis thaliana* miR159b as template backbone (Figure 1A, Appendix A). An overexpression vector for amiR-*NbAGO2* was constructed in CaLCuV (cabbage leaf-curl virus, *Geminivirus*) -based vector pCVA under 35S promoter [37]. The pCVA-amiR-*NbAGO2* was transformed into *Agrobacterium tumefaciens* strain GV3101 and agro-infiltrated into *N. benthamiana* leaves together with pCVB. We used three different regions of *NbAGO2* for amiR targeting, and named them amiR-*NbAGO2-*1, amiR-*NbAGO2-*2 and amiR-*NbAGO2-*3 respectively (Figure 1B). Three days later, silencing effects on *NbAGO2* expression were determined by qRT-PCR experiment. As shown in Figure 1C, overexpression of amiR-*NbAGO2* significantly down-regulated the level of *NbAGO2* by 53–74%. The most pronounced effect was observed for amiR-*NbAGO2-*2. The results demonstrated the efficacy of amiR constructs for knockdown of *NbAGO2* expression, and different amiR varied as regards to silencing efficacy. Therefore, we used amiR-*NbAGO2-*2 for following experiments. As inoculation of pCV interfere with subsequent infection of other viruses like TMV-GFP (Appendix A), we then used pBI121-based binary vector to overexpress amiR. Time course experiment showed that silencing efficiency of pBI121-amiR was most evident at 6 DPI (days post infection) (Figure 2). In the following experiments, we used pBI121-amiR-*NbAGO2-*2 for *NbAGO2* silencing experiments. To verify the effectiveness of amiRNAs on *NbAGO2* expression, another internal control, *GAPDH*, was used and the results were similar to the results from *NbActin* as internal control (Appendix A).

### 3.2. Silencing of *NbAGO2* Compromised dsRNA-Mediated Gene Silencing in N. benthamiana

To confirm that *NbAGO2* is functionally involved in RNAi process, a GFP16c *N. benthamiana* line which overexpresses green fluorescent protein (GFP) was used. First, we used pCB2004B-hpGFP to transiently silence *GFP* expression in GFP16c plant. The result showed that GFP expression was largely silenced by transient overexpression of hpGFP by agroinfiltration, and both GFP fluorescence and *GFP* mRNA levels decreased by hpGFP expression. One should note that GFP signal in GFP16c plant was mostly localized in vascular tissue. Next, we used this system to investigate the involvement of *NbAGO2* in RNA silencing. Down-regulation of *NbAGO2* by amiR-*NbAGO2* recovered both the GFP fluorescence and *GFP* mRNA levels in the hpGFP expressed GFP16c plant, while in the control plant *GFP* was significantly silenced (Figure 3A,B). The loss of silencing by *NbAGO2* knockdown via amiR-*NbAGO2* transient expression verified the participation of *NbAGO2* in RNAi process.

### 3.3. Expression of *NbAGO2* was Induced by Both MeSA Treatment and TMV Infection

Salicylic acid (SA) is reported to be broadly engaged in virus infection and defense response. We also investigated the involvement of SA in the TMV infection process in *N. benthamiana*. Exogenous MeSA was sprayed on three-leaf stage *N. benthamiana* seedlings and the seedlings were collected 2, 10 and 24 h post spray. To verify the effectiveness of SA treatment, expression of SA marker gene *NbPR1a* was determined by Northern blot experiment. As shown in Figure 4A, *NbPR1a* was strongly induced by MeSA treatment after 10 h, demonstrating that foliar application of MeSA is effective in *N. benthamiana*. Quantitative real-time PCR was performed to determine the mRNA level of *NbAGO2*. Under MeSA treatment, transcript level of *NbAGO2* was shortly elevated at 10 h by 2.4-fold and then declined to basal level by 24 h (Figure 4B). We then evaluated the expression of *NbAGO2* in response to TMV infection. *NbAGO2* was significantly induced by TMV infection during 1–4 DPI (Figure 5A). In tobacco, SA was responsible for plant resistance to virus infection, especially of TMV. Our data showed that SA level was significantly elevated by TMV infection (Figure 5B), and *NbPR1a* was also induced by TMV infection in *N. benthamiana* (Figure 4A).

### 3.4. Down-Regulation of *NbAGO2* Promoted TMV Infection

As shown above, *NbAGO2* was responsive to TMV infection and MeSA treatment. Involvement of *NbAGO2* in TMV infection was further investigated as follows. *NbAGO2* was first silenced by pBI121-amiR-*NbAGO2* agroinfiltration, and after one day the same leaf region was agroinfiltrated with TMV-GFP, an infectious TMV engineered to express green fluorescent protein, which was retained in the infiltrated area for at least 6 days. The inoculated leaves were monitored daily for the appearance of GFP fluorescence. As shown in Figure 6A, silencing of *NbAGO2* caused stronger accumulation of virus (GFP fluorescence) as compared to control empty vector (pBI121) inoculated leaves. In accordance with this, overexpression of *NbAGO2* (Ox-*NbAGO2*) resulted in weaker virus accumulation, and this effect was abolished by simultaneous amiR-*NbAGO2* expression. In addition, virus spread was reduced by treatment with MeSA (Figure 6B). TMV RNA level was also determined by Northern blot experiment with CP (Capsid protein) region-specific probe, and the result was consistent with GFP fluorescence data (Figure 6C,D). Taken together, we concluded that *NbAGO2* was an antiviral factor effective against TMV, and SA was implicated in this process by inducing *NbAGO2* expression, at least to some extent.

### 3.5. STTM_miR403a Suppressed TMV Infection

As a core element in RISC formation, *AGO2* plays an important role in RNAi-mediated disease resistance response. We therefore hypothesized that some miRNA, especially which participate in RNAi, may have an effect in the process. We chose three miRNAs to further verify this. Among them, miRNA390a and miRNA393a were reported to influence the activity of *AGO2* by binding to it [41], while miRNA403a was reported to mitigate *AGO2* expression by posttranscriptional regulation [42]. Short tandem target mimic (STTM) is a complementary RNA which blocks small RNA functions in plants by sponge activity. Therefore, we designed an STTM inhibitory vector for each miRNA to examine functions of these miRNAs in virus defense according to the methods described by Tang et al. [43]. After co-inoculation with STTM constructs and TMV-GFP via agroinfiltration in *N. benthamiana* plants, the GFP fluorescence was observably weakened in STTM_miR390a, STTM_miR393a, and STTM_miR403a co-infected area at 6 DPI compared to pGreen_GUS_competitor, which served as the control construct (Figure 7A). Subsequently, fluorescence intensity was quantified, and the GFP intensity of STTM co-expressed groups, especially that of STTM_miR403a, showed statistical differences compared with the control (Figure 7B). In the subsequent analysis, we found that miR403a had a target site in 3′ UTR of *NbAGO2* transcript (Appendix A). STTM_miR403a expression reduced *NbAGO2* mRNA level (Figure 7C), while STTM_miR390a, STTM_miR393a expression did not alter *NbAGO2* mRNA level. Our data showed the regulatory role of miR403a on the expression of *NbAGO2* as well as its role in antiviral response.

## 4. Discussion

SA-mediated defenses were involved in plant response against a variety of pathogen types including viruses, bacteria and fungi [44]. Involvement of SA in plant response against virus infection was first established for the well-known interaction between TMV and its resistance gene *N* from tobacco [45]. Furthermore, SA was reported to be inductively accumulated after virus infection, highlighting the importance of this plant hormone in compatible plant–virus interactions [46,47]. Both RNA viruses and DNA viruses were reported to trigger SA signaling pathway [48,49]. In fact, accumulation of SA after pathogen attack and induction of hallmark *PR* genes were commonly observed [50,51]. Infection of a diverse range of plant pathogens activated SA signaling pathway and subsequent hypersensitive reaction (HR) or systemic acquired resistance (SAR) [51]. RNAi is a well-known antiviral defense mechanism in plants. Cross-talk of RNA silencing and SA triggered defense responses were manifested by several reports. First, *N. benthamiana* AGO4 protein was involved in translational repression and was indispensable for SA-mediated defense response by *N* gene [17]. Second, silencing suppressor proteins encoded by virus genome influence both RNAi and SA-induced plant defenses against viruses [52,53,54]. Third, expression of some RNA silencing component genes are influenced by SA treatment. For example, among plant RNA silencing components, activity or expression level of RdRp1 from a variety of plant species including *Arabidopsis thaliana*, *N. benthamiana*, maize, cotton, potato and tobacco were elicited by SA or virus infection, as led to activation of antiviral defense responses [26,27,55,56,57,58,59]. Our data showed that foliar application of MeSA induced *NbAGO2* expression and activated defense against TMV, as is consistent with results from other studies [60,61]. We used TMV infectious clone pJL24 and it was retained in the infiltrated area within our detection time. Induction of *NbAGO2* expression by SA indicated that cross-talk of RNA silencing and SA-mediated antiviral defenses were mediated by RdRp or AGO2 in some plant species like *N. benthamiana* [62]. As for the TMV defense reaction, *NbRdRp1* was reported to function in systemic leaves but not on directly-inoculated leaves [27,55,63], while *NbAGO2* played TMV defense role in inoculated leaves in our results.

Induction of RNAi pathway component genes by corresponding pathogens were reported in several manuscripts. In *Arabidopsis thaliana*, *AGO2* was induced by bacterial pathogen *Pseudomonas syringae*, virus pathogens TCV, CMV and also by virus silencing suppressor proteins [41,53,64]. *RdRp1* expression was induced by TMV infection [55]. Expression of *DRB4* was induced by infection of viruses such as tobacco yellow mosaic virus (TYMV, *Tymovirus*), TCV, CaMV [65] and TCV [66,67]. In rice, RSV infection induced expression of AGO18 and AGO2, while RDV infection induced expression of several *OsRDR* genes but not *OsAGO* genes [68]. Here, we showed that expression of *NbAGO2* was induced by TMV infection in virological model species *N. benthamiana*. *NbAGO2* was shown to be involved in both RNAi and virus defense responses here. *NbAGO2* was reported to mediate defense against tomato bushy stunt virus (TBSV, *Tombusvirus*) [32]. CRISPR/Cas9 mediated inactivation of *NbAGO2* compromise defenses against PVX, TuMV and TCV, but not *Tombusvirus* and CMV infections [33]. Transgenic approach showed that AGO2 defends against TBSV, TMV and PVX, but not foxtail mosaic virus (FoMV, *Potexvirus*) [69]. Our data add weight to the present evidence that *AGO2* is a broad spectrum antiviral gene whose activity is achieved by RNAi mechanism. Artificial miRNA was shown to be effective to silence variety of genes in a diversity of plant species including *N. benthamiana* [70], *Arabidopsis thaliana* [71], *N. tabacum* [72], *Oryza sativa* [35], *Glycine max* [73], *Solanum melongena* [74], *Medicago truncutula* [75], *Triticum aestivum* [76], *Vitis vinifera* [77], *Solanum tuberosum* [78] and *Solanum lycopersicum* [79]. Here, we also showed that artificial miRNA was an effective way to silence *N. benthamiana* AGO family genes via transient expression from both virus vector and binary vector. As infection of virus vector (pCV) hindered subsequent TMV infection, we used pBI121 for expression of amiRNA and its effects on TMV infection. It is interesting to note that *AGO2* itself is targeted by miR403a in plants [80], and here we showed for the first time that attenuation of miR403a inhibition on *NbAGO2* by STTM technology influenced TMV infection. As for the miR390a and miR393a, both of them were reported to bind to *NbAGO2* protein competitively [41], therefore we speculated that this is the reason that they play similar role with miR403a in our experiments. Paudel et al. reported that *NbAGO2* expression was induced by tomato ringspot virus (ToRSV; *Secoviridae*) isolates ToRSV-Rasp1 or ToRSV-GYV, and silencing of *NbAGO2* elevated ToRSV accumulation [81]. This indicated that *NbAGO2* was involved in plant responses toward different families of viruses. 

## 5. Conclusions

The role of *NbAGO2* gene against TMV infection was investigated in *N. benthamiana*. *NbAGO2* was demonstrated to be involved in RNAi process in *N. benthamiana*. *NbAGO2* mRNA level was elevated by both TMV infection and MeSA treatment. Suppression of *NbAGO2* gene by amiR technology compromised plant resistance against TMV infection. Inhibition of endogenous miR403a, a predicted regulatory microRNA of *NbAGO2*, reduced TMV infection. Taken all this together, we conclude that SA participates in *NbAGO2* mediated TMV response and miR403a is also involved in this response by post-transcriptionally regulating *NbAGO2* expression (Figure 8).

## Figures and Tables

**Figure 1 genes-10-00526-f001:**
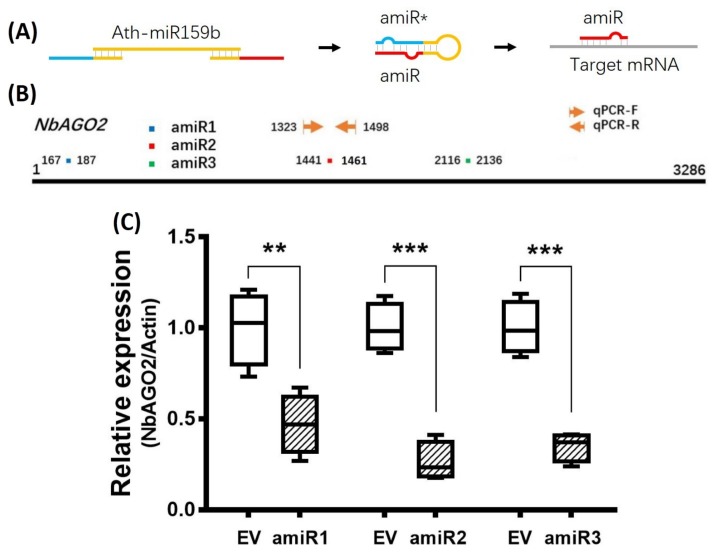
Silencing of *NbAGO2* via transient expression of different amiRs from pCV-based vector. (**A**) Oligonucleotides for amiR was designed, and the sequences were used as primers to amplify *Arabidopsis thaliana* miR159b to produce amiR construct. After transcription and processing, mature amiR binds to complementary site at target mRNA. amiR* denotes amiR star strand. (**B**) The positions of amiR target sequences on *NbAGO2* were shown with different colors, and qPCR primers for detection of *NbAGO2* expression were shown with arrowhead. The numbers indicated the positions of amiR target sites and qPCR primers. (**C**) Different amiR constructs cloned in pCVA vector were transiently expressed in *N. benthamiana* leaves by agroinfiltration. At 3 DPI, plants were harvested and were assayed for *NbAGO2* expression by quantitative real-time PCR. Plants inoculated with empty pCV vector (EV) served as control treatment, and *NbAGO2* expression in control group was arbitrarily designated as 1. *N. benthamiana NbActin* gene served as the internal control. Significant differences between control and treatment groups were calculated by Student’s *t*-test. Double and three asterisks indicated significant difference at *p* < 0.01 and *p* < 0.001, respectively.

**Figure 2 genes-10-00526-f002:**
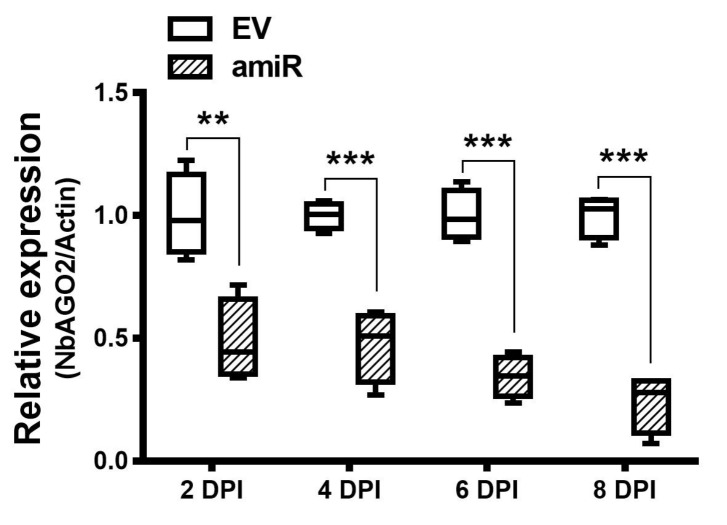
Silencing of *NbAGO2* via transient expression of amiR2 from PBI121-based vector. pBI121-amiR2 construct was transformed into GV3101, infiltrated on *N. benthamiana* leaves, and the leaves were collected at different DPI. *NbAGO2* expression was determined by quantitative real-time PCR. Plants inoculated with empty pBI121 vector (EV) served as control treatment. *NbAGO2* expression in control was arbitrarily designated as 1. *NbActin* gene served as the internal control. Significant differences between control and treatment groups were calculated by Student’s *t*-test. Double and three asterisks indicated significant difference at *p* < 0.01 and *p* < 0.001, respectively.

**Figure 3 genes-10-00526-f003:**
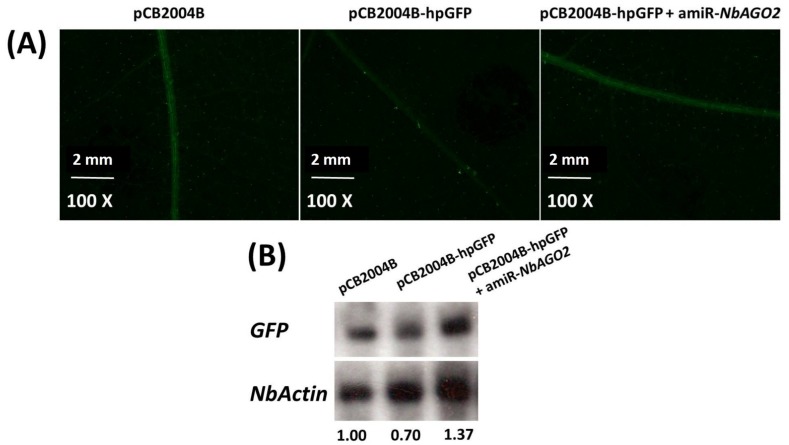
Silencing of *NbAGO2* compromised hairpin induced RNAi in *N. benthamiana*. (**A**) Three-leaf stage GFP16c transgenic plants were inoculated with different combinations of vector constructs. GFP fluorescence was visualized under Nikon SMZ18 microscope at 3 DPI. (**B**) The infiltrated leaf tissues were harvested, total RNAs were extracted, and *GFP* mRNA levels were determined by Northern blot experiment. Expression of *N. benthamian*a β-actin served as internal control. The number below the panel indicates the ratio of the blot band intensity between *GFP* and *NbActin*, in which the ratio in first lane was arbitrarily designated as 1.

**Figure 4 genes-10-00526-f004:**
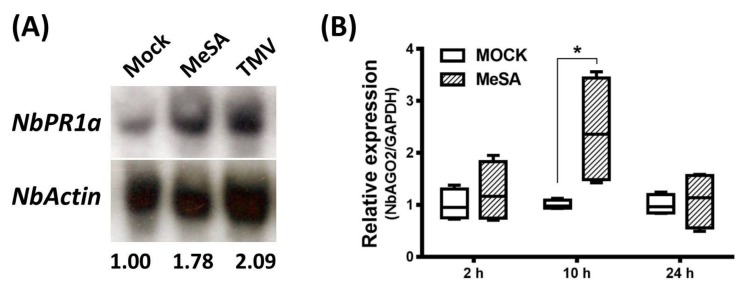
Exogenous spray of MeSA induced *NbAGO2* expression. (**A**) 500 µM of MeSA was sprayed on *N. benthamiana* leaves and the leaves were collected at 10 h post spray. Total RNAs were isolated and *NbPR1a* level was determined by Northern blot experiment using *NbPR1a* sequence specific probe. The same blot also includes the TMV infected leaf sample after 10 h of infection. *NbActin* served as internal control and its mRNA level was evaluated by same way using *NbActin* specific probe. The number below the panel indicates the ratio of the blot band intensity between *NbPR1a* and *NbActin*, in which the ratio in first lane was arbitrarily defined as 1. (**B**) The same leaf tissues treated by MeSA were collected at different time points and *NbAGO2* expression was determined by quantitative real-time PCR. Plants sprayed with water solution served as control treatment. *NbAGO2* expression in control was arbitrarily designated as 1. *N. benthamiana GAPDH* gene served as the internal control. Significant differences between groups were calculated by Student’s *t*-test. Single asterisk indicated significant difference at *p* < 0.05.

**Figure 5 genes-10-00526-f005:**
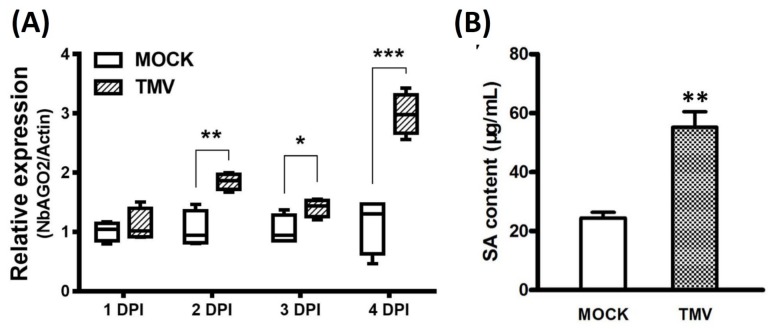
TMV infection elevated *NbAGO2* expression and endogenous SA level. (**A**) Three-leaf stage *N. benthamiana* was infected with TMV, and the leaves were collected at different DPI. *NbAGO2* expression was determined by quantitative real-time PCR. Plants inoculated with inoculation buffer containing empty vector expression GV3101 served as control treatment. *NbAGO2* expression in control was arbitrarily designated as 1. *NbActin* gene served as the internal control. (**B**) Three-leaf stage *N. benthamiana* was infected with TMV, and the leaves were collected 5 h post infection. SA level was determined by UV Vis Spectrophotometry at 310 nm. Significant differences between mock and infected groups were calculated by Student’s *t*-test. Single, double, and three asterisks indicated significant difference at *p* < 0.05, *p* < 0.01, and *p* < 0.001, respectively.

**Figure 6 genes-10-00526-f006:**
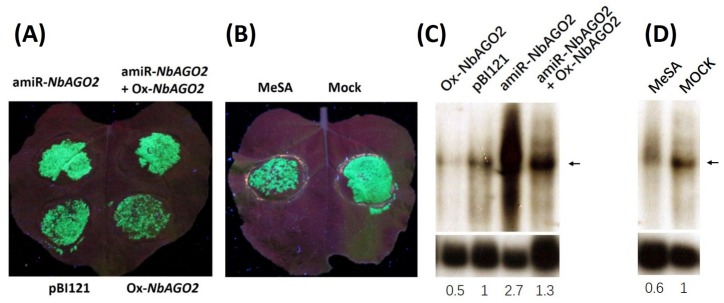
*NbAGO2* contributed to TMV defense in *N. benthamiana*. (**A**) *N. benthamiana* plants were first inoculated with different combinations of vector constructs by agroinfiltration, and infected with TMV-GFP after one day of inoculation. GFP fluorescence was visualized by hand-held UV lamp at 320 nm, and the photographs were taken five days post TMV infection. pBI121, empty pBI121 vector; amiR-*NbAGO2*, amiR-*NbAGO2* construct in pBI121 vector; Ox-*NbAGO2*, *NbAGO2* overexpression construct in pBI121. (**B**) *N. benthamiana* plants were sprayed with 500 µM MeSA daily, and infected with TMV-GFP after the first spray. GFP fluorescence was visualized by hand-held UV lamp at 320 nm, and the photographs were taken five days post TMV infection. (**C**,**D**) Detection of TMV genomic RNA by Northern blot experiment. The same treated samples from above were subjected to detection of TMV RNA. Total RNAs were isolated and TMV RNA level was determined by Northern blot experiment using capsid protein sequence specific probe labelled with ^32^P. *NbActin* served as internal control and the mRNA level of *NbActin* was evaluated in the same way using ^32^P labelled *NbActin* specific probe (lower panel). The band corresponding to TMV genomic DNA was indicated by the arrowhead. The number below the panel indicates the ratio of the blot band intensity between TMV and *NbActin*, in which the ratio in control lane was arbitrarily designated as 1.

**Figure 7 genes-10-00526-f007:**
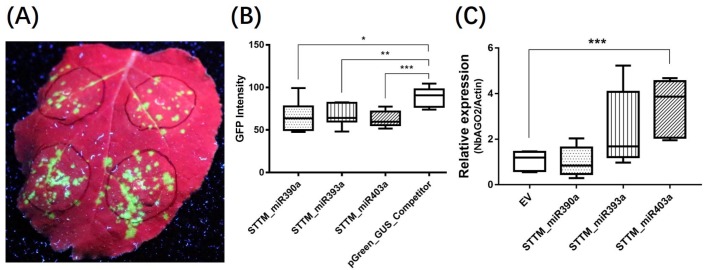
STTM_miR403a suppressed TMV infection. (**A**) *N. benthamiana* plants were co-inoculated with vector constructs and TMV-GFP by agroinfiltration. GFP fluorescence was visualized by hand-held UV lamp at 320 nm, and the photographs were taken at 6 DPI. STTM_miR390a, STTM_miR393a, and STTM_miR403a constructs were cloned in pGreen_GUS_Competitor vector. (**B**) The GFP fluorescence of infection area in (**A**) was converted into gray density by Gel-Pro Analyzer software (V4.0, Media Cybernetics, Inc., Rockville, USA), Dunnett’s multiple comparisons test was used for comparison of GFP intensity value of seven infected leaves. (**C**) Total RNAs were extracted from the same leaf regions from above experiments, and *NbAGO2* expression was determined by qRT-PCR experiment using *NbActin* mRNA level as internal control. Single, double, and three asterisks indicated significant difference at *p* < 0.05, *p* < 0.01, and *p* < 0.001, respectively.

**Figure 8 genes-10-00526-f008:**
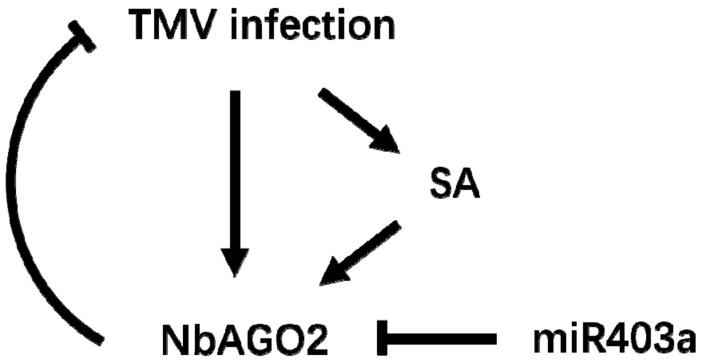
Model for *NbAGO2* mediated antiviral defenses against TMV infection in *Nicotiana benthamiana*.

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
