# Peer review of "miR403a and SA Are Involved in NbAGO2 Mediated Antiviral Defenses Against TMV Infection in Nicotiana benthamiana"

_genes, 2019, doi:10.3390/genes10070526_

Round 1
Reviewer 1 Report
This manuscript described the work done to identify the implication of mi RNA and salicylic acid in Nicotiana benthamiana plant defense mechanisms mediated by AGO2. Protocols are well described and results conclusive. I have only few editorial corrections (see attached file).

Author Response
Reviewer 1: This manuscript described the work done to identify the implication of miRNA and salicylic acid in Nicotiana benthamiana plant defense mechanisms mediated by AGO2. Protocols are well described and results conclusive. I have only few editorial corrections (see attached file)
Answer: Thanks for the reviewer’s kind suggestion. We accepted all of the reviewer’s suggestion and revised the manuscript. Please refer to the revised version on page 1, lanes 30; page 2, lanes 78, 82; page 9, Figure 6 Legend; page 10, Figure 7 Legend; and in the Discussion on pages 10-11.

Reviewer 2 Report
Review report about Diao et al „miR403a and SA are Involved in NbAGO2 Mediated 2 Antiviral Defences against TMV Infection in 3 Nicotiana benthamiana”.
In this work Diao and colleagues investigates the role of AGO2 in N.benthamiana during TMV infection. They use all comprehensive methods for altering AGO2 level – in a transient system- and investigate its consequence. They tried to correlate AGO2 level with SA and miR403a regulation, why at the end they were able to hypothesize a model about this regulation.
I think that the topic is interesting the methods are adequate; most of the results are sound why I think this paper would be good to accept for publication in MDPI Genes.
However, I have some suggestions and some advised experiments, which would increase the value of the paper, why I suggest major revision. If the experiments cannot be done during the short revision period I would rejected the paper highlighting that I highly recommend resubmission when the requested experiments were carried out.
Main problems:
As a final conclusion of the results a model was generated, but I think some further experiments would need to be able to drawn it.
It was shown that TMV infection induced SA and AGO2 level, SA induced AGO2 level, but it was not investigated how the mir403a STTM was affected AGO2 level. I think that investigation of the AGO2 level in this experiments is crucial.
Beside this I have some minor points which would be nice to be corrected in the revised version:
1/ In the abstract it should be written that silencing of AGO2 was carried out in a transient system. Results of transient expressions could differed from the ones when investigations are carried out in stable transformants and virus infection follows this and systemic leaves are investigated.
line40:RNAi was not find in plant virus, but in plants during virus infection.
2/ in the introduction from line53-line70 all results are collected where key regulators of RNA played important role during a particular virus infection. “They mediate defence” or “defend” expressions are used. I think they not defend, but juts have key role during virus infection why I would suggest changing these expressions – it could be misleading.
3/In the Materials and methods section: It is very difficult to follow what constructs were prepared. I would use subtopics in what I would place a sentence about what constructs was produced for what purpose, and describe the primers only after that: hp construct for GFP, amiR constructs for AGO2 (both viral and not viral), overexpression of AGO2 and STTM vectors for miRNAs.
4/ line 169: please indicate what type of membrane was used.
5/line 171: part of the investigated genes, which was used to prepare for radioactive probe hybridization more details would needed. You can describe the primers, which was used for template preparation, or give the sequence of the template as a supplementary material.
6/ 3.1 results as a start it would be good to have a paragraph about amiR AGO2 target design (the sketch on the Fig1 I think is not given enough details). It would be nice to have alignment from different AGO2s (originated from different species) at least covering the part of amiR targeted parts.
7/ Instead of column diagram I would suggest to show qRT-PCR results as whisker diagrams.
8/ On Fig2/A results of the semiquantitative RT-PCR for AGO2 is not clearly seen why I would suggest leaving out this part of the Figure.
9/ line 205: AGO2 has not positive effect, but it influenced the GFP silencing.
10/ During TMV infection AGO2 level was verified by qRT-PCR using GAPDH as an internal control. GAPDH is massively down-regulated in TMV infected plants (Havelda et al 2008 https://doi.org/10.1111/j.1365-313X.2008.03501.x). At the beginning of this manuscript it was shown that AGO2 level in amiR silenced plants were measured by qRT-PCR using either GAPDH or actin. I would suggest to use actin – always during virus infection, because GAPDH as a reference would generate misleading results (Olah et al 2016, doi: 10.1007/s00705-016-2921-9.). Is is very possibly that this intense increase in AGO2 level was calculated because expression of GAPDH (the reference which levels should be constant) was decreased. Therefore, I would suggest repeating this analysis using actin as an internal control.
11/ In the legend of Fig 5, instead of “with inoculation buffer with GV3101”, “with inoculation buffer containing empty vector expression GV3101” should be written – at least this is the correct negative control.
12/ As TMV infection was done by infiltration not with in vitro transcript inoculation, the dimensions of the infiltrated patches are different from one infection to the other why conclusion for virus spread cannot conclude (line 232).
13/ On Fig6 for easy understanding the presence of TMV-GFP should be indicated in all infiltration. Otherwise, it is difficult to see why GFP is present in these leaves. Also on the panel C it should be indicated what probes were used for the hybridizations (it is not enough to write it down in the legend). Mock on B panel means that it was sprayed with water and this is also misleading. I would separate the Northern for MeSA with this mock and the other experiment for easy understanding.
14/ 3.5 at the beginning a paragraph is needed about why these miRNAs were selected in more details. First appearance STTM should be described.
15/ Infiltration on Fig7 seems a bit strange with small patches. Here at least investigation the AGO2 level is essential (As I have already described at the beginning).
16/ It is a well known that miR403a targets AGO2, it should be describe at he beginning of 3.5.
17/ the statement (line 285) is too early. It cannot be concluded only because of the slight change in the GFP fluorescence.
I would suggest to the authors to read and also compare theire results with:
“Expression and antiviral function of ARGONAUTE 2 in Nicotiana benthamiana plants infected with two isolates of tomato ringspot virus with varying degrees of virulence.” by Paudel – as it addresses a quite similar problem (doi: 10.1016/j.virol.2018.08.016.).
As a summary, I think that this manuscript would be good to be accepted in MDPI Genes, but before that revision, addressing my main and minor concerns should be addressed and corrected.
Author Response
Reviewer 2:
Our reply: Overall, the comments pointed out very important issues, especially the requirement of verification of NbAGO expression after STTM expression and detail on experimental design. We really appreciated the comments and incorporated them into the revised manuscript.
Point 1: It was shown that TMV infection induced SA and AGO2 level, SA induced AGO2 level, but it was not investigated how the mir403a STTM was affected AGO2 level. I think that investigation of the AGO2 level in this experiments is crucial.
Answer: Done. We did this experiment and the result was shown as Figure 7(C). Please refer to the revised version on page 10.
Point 2: 1/ In the abstract it should be written that silencing of AGO2 was carried out in a transient system. Results of transient expressions could differed from the ones when investigations are carried out in stable transformants and virus infection follows this and systemic leaves are investigated.
Answer: Done. Please refer to the revised version on page 1 lines 17-21.
Point 3: line40: RNAi was not find in plant virus, but in plants during virus infection.
Answer: Corrected. Please refer to the revised version on page 1 line 41.
Point 4: 2/ in the introduction from line53-line70 all results are collected where key regulators of RNA played important role during a particular virus infection. “They mediate defence” or “defend” expressions are used. I think they not defend, but juts have key role during virus infection why I would suggest changing these expressions – it could be misleading.
Answer: Corrected. Please refer to the revised version on page 2 lines 54-71.
Point 5: 3/In the Materials and methods section: It is very difficult to follow what constructs were prepared. I would use subtopics in what I would place a sentence about what constructs was produced for what purpose, and describe the primers only after that: hp construct for GFP, amiR constructs for AGO2 (both viral and not viral), overexpression of AGO2 and STTM vectors for miRNAs.
Answer: Done. We provided the details on the vector construction and divided different vectors by subtopics. Please refer to the revised version on pages 3-4 lines 106-144.
Point 6: 4/ line 169: please indicate what type of membrane was used.
Answer: Done. Please refer to the revised version on page 4 line 175.
Point 7: 5/line 171: part of the investigated genes, which was used to prepare for radioactive probe hybridization more details would needed. You can describe the primers, which was used for template preparation, or give the sequence of the template as a supplementary material.
Answer: Done. The primers for probe preparation were provided in Materials and Methods. Please refer to the revised version on pages 4-5 lines 178-185.
Point 8: 6/ 3.1 results as a start it would be good to have a paragraph about amiR AGO2 target design (the sketch on the Fig1 I think is not given enough details). It would be nice to have alignment from different AGO2s (originated from different species) at least covering the part of amiR targeted parts.
Answer: Done. More detail about amiR design for target AGO2 was provided in Materials and Methods section on page 3, lines 112-127; Figure 1(A and B) and the Legend on page 5; main text on page 5, lines 188-191. The alignment result was provided as Supplementary Figure 1.
Point 9: 7/ Instead of column diagram I would suggest to show qRT-PCR results as whisker diagrams.
Answer: Done. qRT-PCR results were all provided as whisker diagrams. Please refer to the revised version for Figure 1C, Figure 2, Figure 4B, Figure 5A, and Figure 7C on pages 5, 6, 7, 8 and 10 respectively.
Point 10: 8/ On Fig2/A results of the semiquantitative RT-PCR for AGO2 is not clearly seen why I would suggest leaving out this part of the Figure.
Answer: Done. The semiquantitative RT-PCR result was omitted. Please refer to the revised version for Figure 2 on page 6.
Point 11: 9/ line 205: AGO2 has not positive effect, but it influenced the GFP silencing.
Answer: Corrected. Please refer to the revised version on page 6 lines 217-219.
Point 12: 10/ During TMV infection AGO2 level was verified by qRT-PCR using GAPDH as an internal control. GAPDH is massively down-regulated in TMV infected plants (Havelda et al 2008 https://doi.org/10.1111/j.1365-313X.2008.03501.x). At the beginning of this manuscript it was shown that AGO2 level in amiR silenced plants were measured by qRT-PCR using either GAPDH or actin. I would suggest to use actin – always during virus infection, because GAPDH as a reference would generate misleading results (Olah et al 2016, doi: 10.1007/s00705-016-2921-9.). Is is very possibly that this intense increase in AGO2 level was calculated because expression of GAPDH (the reference which levels should be constant) was decreased. Therefore, I would suggest repeating this analysis using actin as an internal control.
Answer: Done. We did the experiment using NbActin as internal control. Please refer to the revised version Figure 5(A) on page 8. Actually, we replaced most of the qPCR experiment by NbActin as internal control in this revision. Please note this for review.
Point 13: 11/ In the legend of Fig 5, instead of “with inoculation buffer with GV3101”, “with inoculation buffer containing empty vector expression GV3101” should be written – at least this is the correct negative control.
Answer: Corrected. Please refer to the revised version on page 8 Figure 5 Legend.
Point 14: 12/ As TMV infection was done by infiltration not with in vitro transcript inoculation, the dimensions of the infiltrated patches are different from one infection to the other why conclusion for virus spread cannot conclude (line 232).
Answer: Yes, the reviewer pointed out an important issue. So, our Northern blot experiment in Figure 6(C,D) was carried out to verify it as it used the internal control NbActin for normalization. We also quantified the band density and the result was provided in this revision on page 9.
Point 15: 13/ On Fig6 for easy understanding the presence of TMV-GFP should be indicated in all infiltration. Otherwise, it is difficult to see why GFP is present in these leaves. Also on the panel C it should be indicated what probes were used for the hybridizations (it is not enough to write it down in the legend). Mock on B panel means that it was sprayed with water and this is also misleading. I would separate the Northern for MeSA with this mock and the other experiment for easy understanding.
Answer: Done. The probe information detail was provided in the “Materials and Methods” section on pages 4-5 lines 176-185 and Figure 6 legend on page 9. We separated Figure 6(C) to Figure 6 (C) and (D) for easy understanding.
Point 16: 14/ 3.5 at the beginning a paragraph is needed about why these miRNAs were selected in more details. First appearance STTM should be described.
Answer: Done. Please refer to the revised version on page 9 lines 254-260.
Point 17: 15/ Infiltration on Fig7 seems a bit strange with small patches. Here at least investigation the AGO2 level is essential (As I have already described at the beginning).
Answer: Done. NbAGO2 level was determined and the result was provided as Figure 7(C). Please refer to the revised version on page 10.
Point 18: 16/ It is a well known that miR403a targets AGO2, it should be describe at he beginning of 3.5.
Answer: Done. Please refer to the revised version on page 9 line 256-257.
Point 19: 17/ the statement (line 285) is too early. It cannot be concluded only because of the slight change in the GFP fluorescence.
Answer: Corrected. Please refer to the revised version on page 11 lines 321-325.
Point 20: I would suggest to the authors to read and also compare theire results with:
“Expression and antiviral function of ARGONAUTE 2 in Nicotiana benthamiana plants infected with two isolates of tomato ringspot virus with varying degrees of virulence.” by Paudel – as it addresses a quite similar problem (doi: 10.1016/j.virol.2018.08.016.).
Answer: Done. Please refer to the revised version on page 11 lines 325-328.

Round 2
Reviewer 2 Report
Many thanks for the effort of the authors to answer to my questions and incorporate all of the suggestions what I made. I do think that this revision significantly improved the quality of the paper and I suggest to accept this manuscrip for publication in the MDPI genes.